# The Sustainable Human Resource Practices and Employee Outcomes Link: An HR Process Lens

**Aline Elias \*, Karin Sanders \* and Jing Hu**

School of Management & Governance, University of New South Wales, Kensington, NSW 2052, Australia
\* Correspondence: aline.elias@unsw.edu.au (A.E.); k.sanders@unsw.edu.au (K.S.)

**Abstract:** Sustainable human resource (HR) practices, such as diversity and inclusion, have gained considerable attention in HR research. However, to this point, most of the research has focused on the content of HR practices, rarely considering the HR process perspective. Consequently, the processes that explain the relationships between sustainable HR practices and subsequent employee behavioural outcomes are unclear. In this conceptual paper, we propose a revised process model to explain the effects of sustainable HR practices by building on the Strategic HR process model. We contribute to the sustainable HR literature, the HR process research, and the process model by Wright and Nishii in particular. We conclude the conceptual paper by highlighting future research recommendations.

**Keywords:** sustainable human resources; leadership style; sustainability; managers; HRM system strength

## 1. Introduction

Amid the growth and globalisation pressures, violations of human rights, environmental degradation, and financial injustice are increasingly on the rise [1]. As a result, poverty, (gender) inequality, hunger, and other grand challenges are straining humanity, and threatening the sustainable future of the planet [2]. The emergence of sustainable development, defined as "a development that meets the needs of the present without compromising the ability of future generations to meet their own needs" [3] (p. 41), was initiated by the United Nations (UN). As a pathway towards sustainable development, the UN established sustainable development goals (SDGs) that comprise of 17 goals and 169 targets [4]. To retain global competitiveness [5], organisations are attempting to be more sustainable by building higher resilience to disruptions [6]. They are progressively responding to the needs of society and the environment [7], while trying to retain global competitiveness [5]. Within organisations, a response to the needs of society and the environment can be accomplished by means of their human resources (HR) practices [8,9].

While the sustainable HR management (HRM) approach is also studied as green [10–14], environmental [15], or socially responsible (SR-)HRM [16–18], for this conceptual paper, we focus on sustainable HRM, which is defined as "the adoption of HRM strategies and practices that enable the achievement of financial, social and ecological goals, with an impact inside and outside of the organization and over a long-term horizon" [8] (p. 90). Sustainable HRM is an overarching approach [19,20], which differs from other sub-types (i.e., SR-HRM and green HRM) that have paid less attention to the collective sustainability dimensions [21,22]. There are three lines of research within the sustainable HRM field: the study of employee result-oriented outcomes and well-being (i.e., job performance, emotional status), employee behaviour (i.e., sustainable employee behaviour), and an effective SR-HRM system [23]. In this study, we focus on the second approach by addressing the sustainable HRM practices–employee behaviour relationship to better understand the trickle-down effect of sustainable HR practices. Empirical studies show a positive association between sustainable HR practices and behavioural outcomes, such as organisational citizenship behaviour [24,25]. Eventually, employees' behaviours reflect on the

success of the business and the organisation's long-term competitiveness [23]. Embedded in sustainable HRM are sustainable HR practices, which communicate important goals and desired employee behaviours from the organisation to the employee [26] and are not only restricted to financial activities [27,28]. In other words, they can be characterised as signals that are interpreted by individual employees, which shape their behaviours (e.g., effort, cooperation) [29]. Examples of sustainable HR practices are sustainable training [30], incentives [31], and sustainable compensation [32]. Sustainable HR practices help organisations in their aims toward sustainable development [28,33]. In essence, sustainable HR practices recognise the critical enabling role of CEOs, middle and line managers, HR professionals, and employees in communicating messages that are distinctive, consistent, and reflect a consensus among decision-makers [34].

There are notable differences between strategic HRM and sustainable HRM. Strategic HRM generally focuses on the economic purpose of HRM and is less concerned with the interests of sustainability [35]. It focuses on the connections between the strategy of an organisation and HRM [36], while sustainable HRM focuses further on internal and external sustainability outcomes [27]. Extant research has achieved a positive association between sustainable HR practices and employee behaviour. For instance, sustainable training is found to be positively related to sustainable awareness and pro-environmental behaviour [30,37]. Research has also demonstrates that both the top management team's sustainable commitment [38] and organisational citizenship behaviour [39] act as intermediaries in the relationship between sustainable HR practices and sustainable performance, which is defined as "the performance of firms in relation to society, economy, and environment in the era of sustainable development" [40] (p. 3).

Despite the valuable insights emerging in the sustainable HRM field [9,27,41], there are nonetheless at least three shortcomings we have identified. The first constraint is that the sustainable HR literature devotes great attention to the content of the practices [34,42], limiting the field's ability to unravel the mechanisms of the relationship between sustainable HR practices and employee behavioural outcomes. As a core part of the HR management literature, the HR process research focuses on how HR practices are communicated, implemented by line managers, and perceived by employees [33,40]. Despite the importance of understanding the sustainable HR process, negligeable studies have explored factors related to sustainable HR processes, such as the role of managers in this regard [43,44]. Additional research is needed on the underlying processes [27] through which sustainable HR practices are translated into employee behaviours [45], as the relationship between HR practices and employee (behavioural) outcomes can only be considered effective when the practices are conveyed to employees effectively in order for them to know what the HR practices are [46].

The second shortcoming is that, even though the employee perception of the HR practices is a central element of the process model by Wright and Nishii [47], growing evidence suggests that employee perception is only one element in the information processing that might help to explain the relationship between sustainable HRM and employee outcomes [48]. According to the information processing theory [39], for individuals to make sense of the received information, they select and organise particular information, then interpret, judge, and attribute this information to a specific meaning [39,40]. Scholars iterate the need to focus on employees' understanding of what the organisation primarily expects from them, also known as the HRM system strength [49]. As part of the HR process research [50], the perceived HRM system strength is defined as the "features that create a strong HR system and need to be present in order for the HR practices to communicate their intended effects and ultimately influence firm performance" [46] (p. 215). To explain the integration of the perceived HRM system strength in the proposed model, we referred to signalling theory, which addresses communication by and also within organisations [51–54]. Guest et al. [54] elaborate on signalling theory [55,56], which identifies the roles of the signaller (sender), the signal (message), and the signal receiver and considers line managers as both communicators and implementers of HR practices. Moreover, Guest et al. [54]

argue that Bowen and Ostroff [57], who focus on the strength and quality of the signal, and Nishii et al. [58] who are mainly concerned with the receivers and how they perceive and interpret the signals, fail to consider the three elements of the signalling process.

Finally, the HR process model by Nishii and Wright [47] represents a framework illustrated by phases, which commence in the adoption of practices until better organisational performance is achieved, which also involves potential influencing factors. While the HR process model by Wright and Nishii [47] incorporates different types of managers (senior, line managers), it does not contemplate line managers' leadership styles, which might shape employees' understanding of line management's implementation of HR practices. It is an unfortunate encounter, as "the leadership literature could provide insights into the conditions under which employee and managerial reports are more or less consistent and congruent with each other" [59] (p. 13). In their paper, Wright and Nishii emphasise that "social information may play a significant role in how individuals perceive and interpret the practices" [47] (p. 106). As employees engage in close interactions with managers and are exposed to their leadership behaviours, their line manager's leadership style can impact the effects of HR implementation [60], such as on sustainable employee behaviours. The social information processing theory [61] posits that individuals depend on information received from members in the workplace as a key means to shape their perceptions of the organisation's practices. Thus, following the same mechanism, line managers' sustainability behaviours may have a critical role in influencing employees' understanding of sustainable HR practices [62].

Given the abovementioned gaps, our aim is to propose a sustainable HR process framework that explains the process through which implemented sustainable HR practices lead to the expected sustainable employee behaviours, offering scholarly contributions to the HR process and sustainable HR literature. First, our conceptual paper incorporates HR process research [50] into the emerging sustainable HR field to help overcome the 'black box' that can hamper the understanding of the mechanisms through which sustainable HR practices can impact behavioural outcomes. Specifically, the HRM process refers to "the manner and activities through which HRM content is enacted" [63] (p. 2388). In an attempt to develop a full representation of the HR–performance relationship [46,54], the study uses the HR process lens to explain the relationship between the implemented sustainable HR practices and employee behaviour. In doing so, we aim to gain and share a better understanding of the relationship between sustainable HR practices, on the one hand, and employee behaviour, on the other.

As a second contribution, the proposed framework incorporates perceived sustainable HRM system strength (i.e., the employees' understanding of sustainable HR practices), which is defined as the characteristics of the sustainable HR practices as an alternative to the employees' perception of the implemented HR practices. We integrate perceived sustainable HRM system strength as an intermediary through which the sustainable HR practices, implemented by line managers as key signallers, initiate employee behaviour. In line with previous literature, stronger HRM systems have a stronger impact on outcome variables, as they send clear signals to employees about the expectations of management [64,65]. Guided by signalling theory [51–53], we further elaborate on the quality and strength of the information (signals) received by employees (signal recipients) as a result of the implemented sustainable HR practices (signalled by managers). Backed by insights from signalling theory, and as iterated by Guest et al. [54], we consider line managers as vital communicators and implementers of sustainable HR practices, and employees as receivers and interpreters of these signals. Accordingly, we build the case that (high) HRM system strength should positively influence sustainable employee behaviours. Therefore, rather than having employees' perception of the HR practices, we propose the perceived HRM system strength as vital to developing a better understanding of the relationship between sustainable HR practices and employee behaviour. The inclusion of sustainable HRM system strength, which can explain the relationship between the implemented sustainable HR practices and employee behaviour, offers insights into whether the information from

the implemented HR practices is being interpreted as intended by management to achieve the desired outcomes.

Lastly, the relationship between the implemented sustainable HR practices and the perceived HRM system strength may be influenced by the social environment [49], which comprises of managers and colleagues [66]. Specifically, leadership can help unearth the effects of the implemented HR practices, which provides a lens to understand the HR–performance relationship [67,68]. We, therefore, propose that a manager's sustainable leadership style, which "reflects an emerging consciousness among people who are choosing to live their lives and lead their organizations in ways that account for their impact on the earth, society, and the health of local and global economies" [69] (p. 26), can strengthen the relationship between the implemented sustainable HR practices and the perceived sustainable HRM system strength. We argue that the sustainable leadership style of the line managers is a critical attribute, which shapes the association between the implemented sustainable HR practices and the perceived sustainable HRM system strength, leading to a sturdier demonstration of sustainable employee behaviours, which is defined as "pro-social and pro-environmental behaviour adopted by employees in support of corporate change for sustainability" [70] (p. 1222). In addition, as prior studies have rarely examined the antecedents of sustainable employee behaviours [71], as well as the processes through which sustainable HR practices (implementation) can influence sustainable employee behaviours [45], we also include this particular employee behavioural outcome in our proposed model. When line managers exhibit a commitment to sustainability, they can assist employees in interpreting the intended information and implementing sustainable employee behaviours.

In the following sections, we start by briefly introducing the literature review, followed by the HR process model [47], which underscore our development of the theoretical model. As propositions can help to theoretically specify and explain the relationship between established concepts and can stem from a systematic literature review [72], we followed this approach to develop the propositions. The propositions focus on the application and elaboration of the process model through the implemented sustainable HR practices, the elaboration of employee perceptions, and the impact of managers' sustainable leadership styles. As the study on the effect of HR practices is a multi-level phenomenon [47], we finally elaborate on how empirical studies can be conducted to examine our suggested multi-level framework. By integrating theories across levels of analysis, we attempt to connect organisational concepts to individual concepts and then link them back to other organisational concepts [47]. We conclude the paper by highlighting implications for future research.

## 2. Literature Review

In a systematic literature review [22], Elias and Hu identified and synthesised 36 empirical papers, published between 2012 and 2021, which address sustainable HR practices and examine the role of managers. The review's systematic approach is aligned with the insights by Xiao and Watson [73] who provide guidelines on the different review approaches, how to plan for a review, the quality criteria, and the importance of idea flexibility. While existing reviews [9,74–78] on sustainable HRM and its sub-fields provide good insights on the overall progress in the literature, Elias and Hu's review [22] of sustainable HRM, which has a distinct emphasis on managers, examines managers' hierarchical levels and leadership styles.

In the review, Elias and Hu [22] identified and screened eligible articles by following the PRISMA (Preferred Reporting Items for Systematic Reviews and Meta-Analyses) four-phase flow [79]. After identifying the included papers, they coded the papers and conducted content analysis [80]. To know more about the potential discrepancies between the concept used by researchers and the definition's content, the authors coded the terms which reflect the sustainable development dimensions (environment, society, economy). The authors used the summative content analysis approach [81], which includes the quantification of

specific words to explore their usage and interpret the embedded knowledge. The authors also utilised MS Excel and NVivo (version 12) [82] to facilitate the continuous comparison of key terms in the construct definitions. Elias and Hu [22] coded the individual sustainable HR practices to know more about the choice and breadth of the HR practices chosen, as opposed to the ignored ones. On the individual HR practices that were studied in the coded articles, most papers focused on training and development under ability-enhancing practices, followed by pay and reward under motivation-enhancing HR practices. For instance, Teixeira et al. [83] focused on environmental training, and Piwowar-Sulej [84] focused on training practices. Information sharing and flexible job design practices have not been studied as much by researchers. Potential reasons could be that job security and flexible job design are associated with the social dimension of sustainable HR [85].

While there may be different hierarchical managers in organisations who could be involved in the HR process, in the proposed model, we specifically focus on line managers as they decide if and how effectively they will implement the HR practices. The vital role that line managers play in the HR process is increasingly gaining attention [51,86,87], especially given that supervisors are considered as "interpretive filters of HRM practices" [57] (pp. 215–216). However, in the review of the sustainable HR literature, which focuses on the role of managers [22], researchers mostly focused on the top-management level followed by the line/supervisory level. Guided by the HR process research, we therefore give attention to line managers. In the next section, we explain the different concepts and then present the propositions leading to the proposed model.

## 3. Theoretical Model

### 3.1. A Sustainable HR Process Model

Nishii and Wright [47] developed a so-called process model to describe the variability in the HR practice implementation process. In their follow-up publication, Nishii and Wright [47] differentiated between *intended HR practices,* which are the HR practices adopted by the senior management in an organisation, *actual HR practices* as the HR practices that are implemented by (line) managers, and *perceived HR practices* that are the HR practices as perceived by employees [47]. Based on the *perceived HR practices,* employees translate the information into *reactions* that might be affective (attitudes), cognitive (knowledge or skill) or behavioural, which will impact *organisational-level outcomes*.

Guided by the HR process literature [88] and extending the abovementioned process model by Wright and Nishii [47], we propose a sustainable HR process model. For this proposed model, we develop research propositions on the relationships between the implemented sustainable HR practices and sustainable employee behaviours by including the perceived sustainable HRM system strength and the sustainable leadership style. The proposed model is depicted in the following figure (see Figure 1).

#### 3.1.1. Implemented Sustainable HR Practices

The starting point for our model includes sustainable HR practices, which are implemented by line managers, and are considered as an antecedent to the subsequent employee reactions. Sustainable HR practices are aligned with the environmental, social, and financial sustainability dimensions of the triple bottom line [89]. That said, we address sustainable HR practices as a tri-dimensional construct [90], rather than focusing on a single concern, which was dominantly identified in the systematic literature review [22].

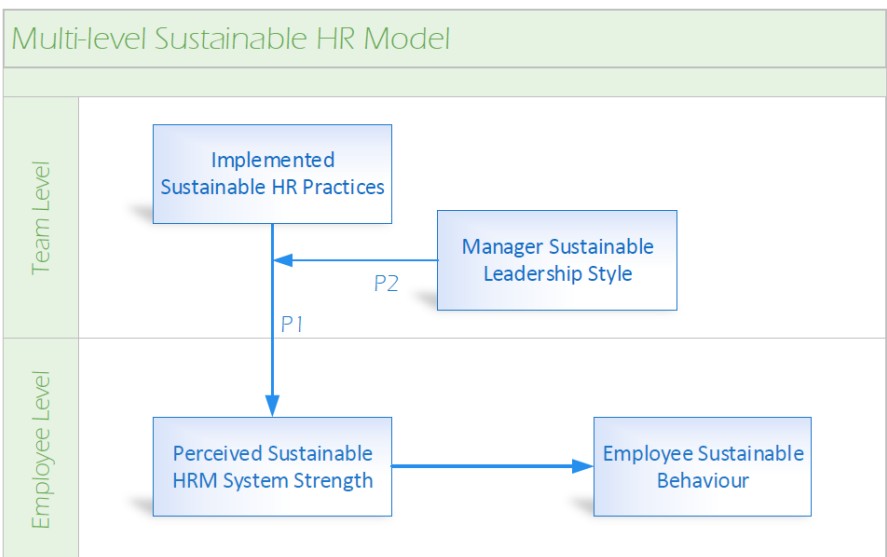

**Figure 1.** Proposed theoretical framework: multi-level sustainable HR process model. Note: P1 as the first research proposition and P2 as the second research proposition.

### 3.1.2. Perceived Sustainable HRM System Strength

In the proposed sustainable HR process model, we include the HRM system strength notion by Bowen and Ostroff [49], which is a stream in the HR process research [50]. Instead of merely giving attention to the content perspective of HR practices [91], their seminal framework focuses on "understanding how HR practices as a system can contribute to firm performance by motivating employees to adopt desired attitudes that, in the collective, help achieve the organization's strategic goals" [46] (p. 204). Ostroff and Bowen [46] suggest that "successful process and implementation of practices rest on ensuring that practices are distinctive and attended to, send consistent messages, and are fair. The net result will be effective and strong HR systems" [46] (p. 201). By applying the co-variation principle of Kelley's attribution theory [92], the HRM system strength concept suggests that HR practices can be considered as persuasion signals from the employer to employees that are aimed at shaping employee reactions [91]. That is, for a message (such as the implemented sustainable HR practices) to have its intended effect (sustainable employee behaviour), the submission, reception, and interpretation of the message is needed [65].

### 3.1.3. Sustainable Employee Behaviour

The dominant sustainable HR literature, which pays attention to the role of managers, is dominantly focussed on the 'environmental' aspect of sustainability [22]. As a result, the literature recognises that employees are vital to a company's environmental efforts [90], rather than just contributing to the overall sustainability scope. However, the role of HRM revolves around "identifying broader purposes for HRM, through its recognition of the complexities of workplace dynamics and the explicit recognition of the need to avoid negative impacts of HRM practices" [34] (p. 1085). Based on the same premise, in the proposed model, we included sustainable employee behaviour as an ultimate outcome. Specifically, to achieve effective workplace sustainability programmes, attention should be directed towards achieving behaviours that aim to minimise the negative impact of an individual's actions on the environment and in the community [93].

### 3.1.4. Sustainable Leadership Style

We build on the literature review [22], described earlier, to identify the focal internal boundary condition in the relationship between the implemented sustainable HR practices and the perceived sustainable HRM system strength. By coding the managers' hierarchical levels and their leadership styles, the review by Elias and Hu [22] found that various

leadership styles of managers were studied in the field of sustainable HR. The results also showed that the ethical leadership style, which emphasises the alignment with what is right and wrong by helping to achieve social welfare and justice [67], was mostly studied. For instance, Ren et al. [38] found that both ethical leadership and green HR were positively associated with top-management team green commitment, which positively impacted environmental performance. Some papers examined green transformational leadership [94,95], which is defined as the "behaviours of leaders who motivate followers to achieve environmental goals and inspire followers to perform beyond expected levels of environmental performance", [96] (p. 109). Others [97,98] broadly studied managers with a transformational leadership style, who are concerned with the transformation or change in followers' fundamental values, goals, and aspirations [99]. In the remaining studies in this review, authors focused on "visionary leadership" as an enabler of sustainable business growth, including that having "growth strategies are usually associated with strong leadership motivation and a superlative performance" [100] (p. 3). In addition, Leidner et al. [11] studied the role of "sustainability advocates", who are "leaders and managers in pursuit of their firm's environmental agenda" [11] (p. 1169), in the preparation and dissemination of environmental HR policies.

While ethical and sustainable leadership are both helpful to achieve organisational success [101], dedication to sustainability would be better off with leaders who could help to understand sustainable practices within organisations [102]. We, consequently, focus on the sustainable leadership style, which "is concerned with creating current and future profits for an organization while improving the lives of all concerned" [103] (p. 209). When followers believe in and follow a sustainable leader, it reflects in their positive behaviour, which ultimately increases their organisational citizenship behaviour [104]. Recent empirical studies indicate that sustainable leadership is a core determinant of long-term success, as well as sustainability performance outcomes [105,106]. According to the upper echelon perspective, the strategies and behaviours of leaders can affect organisational performance or outcomes [107]. In addition, researchers suggest that leadership should go beyond the mere environmental or independent social notions of corporate sustainability and better engage in the triple bottom line [108]. After the above overview of the proposed model's concepts, the following section elaborates on the relationships through the propositions.

### *3.2. Perceived HRM System Strength as an Alternative to Employee Perceptions*

3.2.1. The Effect of Perceived Sustainable HRM System Strength

To explain the relationship between sustainable HR practices and sustainable employee behaviour, we follow Wright and Nishii [47] in their argument that employee perceptions, in addition to the implementation of HR practices by line managers, are central. Employee perceptions of HR practices play an important role in determining the effectiveness of HR practices [52,66]. Kehoe and Wright [109] posit that, employee perceptions are "temporally closer to, and consequently likely to be more predictive of, their attitudinal and behavioural outcomes than are HR practice ratings as provided by managers" (p. 369) when linking HR practices to employee attitudes and behaviours. HR researchers draw a distinction between three approaches in their studies on employee perceptions of HR practices [48,59]. In their review of the studies on employee perceptions of HR practices, Wang et al. [48] distinguish between three forms of perceptions, the what (the content of HR practices), the how (the understanding), and the why (the attributions). Specifically, one line of research focuses on the 'what' of HR practices, which is the content of the HR practices. Another direction gives importance to the 'how' of HR practices that might lead to variable outcomes depending on how employees frame and receive these messages. The last way to examine employee perceptions of HR practices is through the 'why' of HR practices, which is related to the way employees interpret the organisation's motivations underlying the practices [48]. Knowing the mentioned perspectives, we argue that the second approach of perceived sustainable HRM system strength (perceptions of understanding) is more insightful than the perceived content of the sustainable HR practices.

To further uncover the notion of employee perceptions of the implemented sustainable HR practices, we draw on social cognition theory [110], which is centred around how people make sense of their social environment. A potential challenge to HR implementation is that messages might be unclear, complex, or misinterpreted [54], leading to undesired employee behaviours. Based on the information derived from their social context, employees attach different meanings to the social stimuli [111]. In other words, even if managers implement the same HR practices in the organisation, employees will not always interpret it in a similar way. Furthermore, and in the space of sustainable HR, research shows that the perceived HRM system strength is positively related to employee attitudes and behaviours, such as citizenship behaviour [64] and assistance towards co-workers [112]. Similarly, the understanding of goals (such as the SDGs) promotes a shared sense of purpose and aligned thought processes, which improve effective joint working [113]. We, therefore, focus on the perceived HRM system strength rather than the employee perceptions of the implemented HR practices, as the way employees understand HR practices (system) can offer insights on the potential variability between individuals and help avoid detrimental consequences for their behaviour and the organisation [114]. It is, therefore, vital to consider employee perceptions (understanding) of the HR practices system strength, which is yet underexplored in the HR process model by Wright and Nishii [47].

In their elaboration of the HRM system strength concept, Bowen and Ostroff [57] expanded the notion of the sole focus on the content of HR practices. To build their argument, they focused on the importance of the psychological processes, whereby employees attach meanings to communicated HR practices [57]. Bowen and Ostroff [57] followed the covariation principle of the attribution theory [92,115], which helps explain the way individuals process information to make sense of the situation. That is, Ostroff and Bowen [46] argued that when employees perceive HR practices as distinct, consistent, and consensual, they can understand the HR practice as it was intended by management. Consequently, they understand what is expected and rewarded. Adapting to the notion of sustainable HRM system strength, and in alignment with signalling theory [54–56], the implemented sustainable HR practices will relay highly distinct, consistent, and consensual signals to employees that the organisation is committed to sustainable development and expects them to understand and enact such behaviours by demonstrating the expected reactions. Consequently, by exploring the perceived sustainable HRM system strength element as an intermediary, we expect that this will enrich our insights on the relationship between the implemented sustainable HR practices and the expected employee behaviours.

Signalling theory offers an adequate lens in our proposed integration of the two HR process streams by Bowen and Ostroff [57] and Wright and Nishii [47]. Furthermore, social information processing theory [61] acknowledges that social information signals help individuals interpret and subsequently construct meanings, which then influences their attitudes and behaviours. This theory indicates that when employees socially construct their environment, they translate this information into appropriate attitudes and behaviours [61]. Similarly, sustainable HR practices provide information about the organisation's dedication to sustainability [34] to employees who are expected to understand the underlying features of the practices to demonstrate the required behaviours.

In this paper, we specifically suggest that the perceived sustainable HRM system strength will help employees engage further in sustainable employee behaviours. Particularly, researchers have found that employees can contribute towards sustainable development through their behaviour [116,117], which includes pro-social and pro-environmental behaviours. In addition, as there can be detrimental adverse outcomes, employees demonstrating sustainable employee behaviours are also expected to "consciously seek to minimize the negative impact of one's actions on the natural and built world" [93] (p. 240). If employees are not knowledgeable of the expected behaviours, they might fail to demonstrate them [57]. Therefore, guided by information processing theory [110], in alignment with social information processing theory [61], and by integrating the premise of HRM system strength [57], we specifically propose that the perceived sustainable HRM system

strength, which characterises the implemented sustainable HR practices, helps to explain the relationship between the implemented sustainable HR practices and the associated sustainable employee behaviours.

**Proposition 1.** *Employees' perceived sustainable HRM system strength will explain the relationship between sustainable HR practices and sustainable employee behaviours.*

3.2.2. The Impact of Sustainable Leadership Style

Moreover, the relationship between sustainable HR practices and the perceived sustainable HRM system strength might be influenced by the managers' leadership style. As employees engage in close interactions with managers and are exposed to their leadership behaviours, their line manager's leadership style can impact the effect of HR implementation [60] and can influence the sensemaking process of employees [118]. Nishii and Wright [47] highlight that the leadership styles of leaders might affect employee perceptions of HR practices [119]. While the process model by Wright and Nishii [47] focuses on different levels of managers, it overlooks the leadership styles of managers. In their elaboration on the model, Wright and Nishii highlight the role of social information in shaping how individuals perceive and interpret HR practices [47]. Employee perceptions of the HRM system strength are shaped by their social environment [49], which consists of managers as well as colleagues [66]. Specifically, the perceptions and performance of followers are contingent on the leaders' behaviours [120]. That said, through their sustainable leadership style, managers can influence the relationship between the implemented sustainable HR and the perceived HR (system) strength. In contrast, research on abusive supervision implies that highly abusive managers may misuse their power when they are implementing HR practices [63], potentially relaying divergent messages about what is expected from employees. Nevertheless, researchers argue that leadership can help uncover how HR practices are translated to achieve a better understanding of the HR–performance relationship [67,68], such as through their sustainable leadership style which "reflects an emerging consciousness among people who are choosing to live their lives and lead their organizations in ways that account for their impact on the earth, society, and the health of local and global economies" [69] (p. 26).

The sustainable leadership style integrates the perspective of the triple bottom line, encompassing all sustainability dimensions [121] to support the achievement of organisational sustainability [108]. The sustainable leadership style is central when employees are expected to demonstrate behaviours which are aligned with sustainability [27]. Drawing on social information processing theory [61] and the literature on sustainable leadership, we propose that the line manager's sustainable leadership style, which demonstrates ethical and socially responsible behaviours [122,123], can strengthen the positive relationship between the implemented sustainable HR practices and the perceived sustainable HRM system strength which, in turn, helps achieve better sustainable employee behaviours.

**Proposition 2.** *Managers' sustainable leadership style influences the relationship between the implemented sustainable HR practices and the perceived sustainable HRM system strength, such that a better sustainable leadership style strengthens this association.*

## 4. Discussion

In the proposed conceptual model, which provides a foundation for subsequent empirical research, we focus on the umbrella concept of sustainable HR, which is also described as encompassing different sub-types, mainly socially responsible HRM, triple bottom line HRM, and common good HRM [35]. Although the research on sustainable HRM, which extends the strategic HRM literature [124], is evolving and has become more prevalent in the last decade [7,9,42,74], there are ample insights in the HR process literature which can expand our understanding of the sustainable HRM–organisational performance relationship. A HR process view helps to explicate the 'how' in this relationship [64,125].

In this paper, we propose a sustainable HR process model, which is guided by the HR process research. The proposed model identifies a potential intermediary element and an internal contextual factor to explain the adoption of sustainable HR practices. The proposed sustainable HR process model complements the HR process model by Wright and Nishii [47], by focusing on the importance of managers, the leadership style, and sustainable employee behaviours in the sustainable HR field. The proposed model helps to unravel the black box of the sustainable HR–performance relationship by referring to the HR process research. Second, while studies which examine HR system strength as a mediating mechanism are scant [48], the proposed framework explores the concept of perceived sustainable HRM system strength in the sustainable HR process model. Although Bowen and Ostroff's [57] notion of HRM system strength was conceptualised at the level of the organisation [46], our proposed model is aligned with the studies which examine the concept at an individual level by investigating employee perceptions of HRM system strength [64]. We base our emphasis on the perceived HRM system strength, as it was recognised by Ostroff and Bowen [46] as a "meaningful construct" (p. 198). Attention to the HRM system strength can help reveal unexplained variance in the relationships between the implemented HR practices and employee reactions [126]. The study responds to the need to integrate different HR process streams [127] which are the (perceived) HRM system strength and the HR process model by Wright and Nishii [47], by integrating the (perceived) HRM system strength into the proposed sustainable HR process model. As a result, we help to bridge the strategic HRM and sustainable HRM literature by integrating the HR process into the sustainable HRM field.

Lastly, the proposed framework incorporates sustainable leadership as an internal contextual mechanism influencing the relationship between the perceived sustainable HRM system strength and sustainable employee behaviours. As asserted by Johns [128] (p. 584), in regard to disentangling the contextual effects, "if there has been a deficit in contextual theorizing, it is most apparent in a basic lack of theories that treat discrete events as context". Accordingly, we include the sustainable leadership style since sustainable leaders value the different aspects of life over the long run attempting to preserve resources, promote diversity, and focus on enabling learning within an organisation [129]. Capturing the potentially positive effect of the sustainable leadership style of line managers on the relationship between the implemented sustainable HR practices and the perceived sustainable HRM system strength can, therefore, establish new findings in the field of sustainable HR.

### 4.1. Future Directions

Although our proposed model aims to lay the foundation for potential empirical studies, different avenues can be sought to test or extend the propositions. One approach to test the propositions is to conduct longitudinal surveys and experiments. Longitudinal surveys and experiments are valid ways to assess the causality issues between sustainable HR practices and sustainable employee behaviours [28,130–132]. Longitudinal studies can help uncover more information about mechanisms and effects which might vary with time [133], such as the implementation of new sustainable HR practices. Moreover, the process of employees perceiving a strong HRM system can unfold with time, while employees form reactions to the received signals from the HRM system [134]. On the other hand, through an experimental study, and following the approach by Sanders and Yang [135], researchers can study the impact of different sustainable HR practices which might uncover various effects. An example could be that rewards for sustainable employee behaviours would have a stronger impact on sustainable employee behaviours than training on sustainability matters.

In addition, Darvishmotevali and Altinay [136] recommend the utilisation of mixed method approaches, such as through complementing a quantitative study with qualitative interviews. Interviews can be conducted with different parties, such as HR managers,

sustainability experts, managers, and employees, to better understand their motivations towards sustainability.

The proposed model focuses on line managers specifically, however, managers at other levels can be potentially considered [22]. In the sustainable HRM field, managers at different levels help companies to achieve their sustainable goals [137,138]. Researchers can examine the signalling functions at different organisational levels [139], in attempting to uncover potential variability in 'how' signals are interpreted. The interactions and relationships of different managers with employees are critical within organisations [140], therefore it could be helpful to examine the impact of other managers on the relationship between the implemented sustainable HR practices and sustainable employee behaviour.

Scholars highlight the importance of studying the effect of contextual boundary conditions or differences related to the country, such as developing versus developed countries [141]. Other aspects can be considered, such as the effects of cross-cultural differences such as power distance, collectivism, uncertainty avoidance, and performance orientation [142]. Scholars might also elaborate on the impact of individual differences, such as personality, to understand their reactions. Studying the employee's personality helps better comprehend their perceptions of HRM system strength [65].

Another interesting line of inquiry could consider the HRM system strength construct as a higher-order one, as originally intended in the seminal paper by Bowen and Ostroff [57], and further iterated in their follow-up paper [46]. A recent study by Meier-Barthold and colleagues [114] conceptualises and measures HRM system strength as a higher-level characteristic in the HR system, in order to differentiate the feature dimensions from the content of the HR system [64]. Another study [143] assesses the effects of the HRM system strength construct at the individual level, as well as the group levels, which could be interesting to disentangle the potential variability in the proposed sustainable HR process model.

*4.2. Limitations*

Our theoretical model is subject to certain limitations. First, scholars highlight the challenges associated with focusing on the three dimensions of the triple bottom line approach. Researchers suggest that the three-goal focus leads to positive outcomes towards employees, stakeholders, and the environment alike, achieving a 'win–win–win' situation [144,145]. However, when organisations implement sustainable HR practices, there could be underlying dilemmas, paradoxes, and tensions [146], such as making the choice between investing in training an existing employee versus hiring an expert for a new role. Brandl and colleagues [147] draw insights from paradox theory [148], which improves our understanding of the dynamics and tensions among contrasting elements, to introduce an analytical perspective on the response of line managers to competing demands [149]. To illustrate the essence of using a paradox lens, we refer to Newton's physics theory [150]. We describe the paradox perspective as seeing and painting a colour spectrum from a single ray of light. As an example, line managers as paradox navigators [151,152] may experience tensions and resist the deployment of HR practices [153], which could negatively impact employee reactions. However, the proposed model attempts to highlight the crucial role of managers' sustainable leadership styles, as a pre-requisite to balancing the different sustainability dimensions and demonstrating the expected behaviours.

Second, although the seminal model by Wright and Nishii [47] focuses on the importance of examining the variability between the 'intended' and 'actual' practices, as well as the organisational outcomes, our proposed model excludes other linkages in the HR process model. Researchers can take one step further and examine all the linkages to identify potential misalignments between the intended sustainable HR strategy set by the upper management, the implemented practices, the employee reactions, and the organisational outcomes. The implemented HR practices might vary from the intended HR policies or strategies because of differences in the line managers' implementation approaches [52,154]. Examining these relationships can help to uncover how other organisational actors, such as

line managers, implement a practice as it is intended by (senior) managers [155] in their strategy and, eventually, how it impacts their targeted outcomes. As an illustration, the dedication to sustainability is very well captured by deSter company, a leading provider of innovative food packaging and serviceware concepts to the aviation and food service industry. They define their environmental, social, and governance (ESG) strategy as being "committed to creating a positive impact on the world, which doesn't only mean our impact on the environment but also our impact on people and our role in society" [156].

## 5. Conclusions

Our proposed sustainable HR process model unlocks new research horizons by integrating several bodies of knowledge, which are the HR process and sustainable HR. The scholarship around sustainable HR is largely content driven, focusing on the breadth of HR practices rather than how they are perceived by employees. Leading HR process scholars have advanced the strategic HR management literature by uncovering the black box between HR and performance, and by focusing on the underlying mechanisms, which are yet to be unravelled empirically in the sustainable HR literature. Nonetheless, our new research perspective stems from the explicit recognition of the role of line managers and their leadership style, which is aligned with the comprehensive scope of sustainable development. Overall, we provide a connection between sustainable HR practices and the role of line managers in guiding the expected sustainable employee behaviours, attempting to make a difference to the global challenges to sustainable development. Our hope is that HR scholars will continue to develop this line of reasoning to increase our understanding of the process elements and linkages in the proposed HR process model elaborated. By investing more effort in a wider scope of sustainability dimensions, researchers will have a "golden opportunity to make a real difference in the world" [157] (p. 357).

**Author Contributions:** Conceptualization, A.E., K.S. and J.H.; writing—original draft preparation, A.E.; writing—review and editing, A.E., K.S. and J.H. All authors have read and agreed to the published version of the manuscript.

**Funding:** This research received no external funding.

**Institutional Review Board Statement:** Not applicable.

**Informed Consent Statement:** Not applicable.

**Acknowledgments:** We thank the guest editor of this special issue, Roel Schouteten, for his clear guidance throughout the review process, as well as three anonymous reviewers for their constructive feedback and suggestions.

**Conflicts of Interest:** The authors declare no conflict of interest.

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
