# Peer review of "The Sustainable Human Resource Practices and Employee Outcomes Link: An HR Process Lens"

_sustainability, doi:10.3390/su151310124_

Round 1

Reviewer 1 Report

Dear authors, considering that this is a theoretical contribution I believed that there are some aspects that are necessary to reinforce:

1) Introduction and background are well developed. However, I will suggest an strong theoretical reinforcement like systematic literature review made by the author as own contribution; Xiao, Y., & Watson, M. (2019). Guidance on conducting a systematic literature review. Journal of planning education and research39(1), 93-112.

2) Theoretical framework presents a goof basis in relevant articles, however little has been said about the Figure 1. Before entering in section 2.2 I will provide an overview of the model, presenting the variables, and after this I will start with the section 2.2

Despite there are theoretical propositions the article is well written and it is interesting. I will add for practical implications or practitioners work a qualitative analysis based on interviews. After reviewing the study I believe that some questions based on literature review could be used as open questions to know more about the experience of human resource managers in this model. This will be noticeable and great contribution adding more value to the paper.

Author Response

Dear Reviewer,

Thank you for your recommendations. Please find attached our responses.

Best Regards,

Authors

Reviewer 2 Report

Previous research has rarely looked at the causes of employee sustainable behaviors, as well as the methods and procedures by which sustainable HR practices (implementation) can influence employee sustainable behaviors, Thus, the process that explains the relationship between sustainable HR practices and subsequent outcomes is unclear. Thus study contribute to the sustainable HR literature by proposing a revised process model for the effects of sustainable HR practices.

The paper can be accepted with its current form without revisions.

Author Response

Dear Reviewer,

Thank you for your review and recommendations.

Please find attached our response.

Best Regards,

Authors

Reviewer 3 Report

Abstract

The abstract is somewhat confusing on the objective of the research, please clarify.

1. Introduction

Line 67 starts with “Second”, (further occurrences are repeated throughout the body of text), please rearrange the text.

Line 107, same problem mentioned earlier, plus, the type of syntax is repeated.

Line 149, It is advisable to remove “Our model is depicted in Figure 1.” In this part.

2. Theoretical Model

Line 173, …”see also 1” what is this?

Line 225 Proposition 1, is this a research question? A hypothesis? Please clarify

Line 279 Proposition 2, is this a research question? A hypothesis? Please clarify

Line 283, the propositions in the model should be revised (as mentioned in the above lines)

3. Conclusions

Conclusions are very restricted; a suggestion could be the incorporation of limitations and future directions in this part.

Author Response

Dear Reviewer,
Thank you for your review and recommendations.
Please find attached our responses.

Best Regards,
Authors

Round 2

Reviewer 1 Report

It is a very complete work. I believe that after the first review authors improve in a greater extent the previous version of the article. From my perspective the article meet the scientific standards for being published.

Author Response

Thank you for your compliments regarding revised manuscript. We hope that the updated version also resembles your expectations.